# Peer review of "Interpreting Prostate MRI Reports in the Era of Increasing Prostate MRI Utilization: A Urologist’s Perspective"

_diagnostics, 2024, doi:10.3390/diagnostics14101060_

Round 1
Reviewer 1 Report
Comments and Suggestions for Authors
The article is well written and the topic is interesting.
The urologist point of view cocerning MRI reading and interpretation is of great importance to accomplish a more accurate biopsy.
I would suggest the authors to comment the role of CAD and other programs helping radiologist and urologist to read easily MRI images and avoid misunderstanding. Consider reference PMID 33179868.
Author Response
I'm would like to inform you that we have incorporated your suggestion into the manuscript. Specifically, we have cited Ferierro et al.'s research in the chapter "Is it appropriate for radiologists to utilize terms such as "PIRADS 3/4" in their reports, even when these specific PIRADS scores are not explicitly designated?". As you suggested, we have included the relevant details from their study, highlighting the advantages of computational analysis in assisting radiologists with diagnostic accuracy.
Here's the excerpt we added to the manuscript:
"A different approach to addressing this challenge has arisen with the development of computer-aided diagnosis (CAD) systems. Ferierro et al. highlighted the utility of these systems, revealing that radiologists acknowledged the advantages of computational analysis in approximately 15.3% of cases where the PI-RADS Score was ≤3. In the majority of these instances, T2 hypointensity was indistinct, yet a positive Malignancy Attention index map provided by the CAD system assisted radiologists in distinguishing suspicious foci from benign stromal nodules."
Reviewer 2 Report
Comments and Suggestions for Authors
This submission is an interesting Commentary/Perspective piece about mpMRI usage in prostate cancer management. The authors have raised seven questions related to the application of mpMRI and shared their own feedback. Unlike Review Articles, Commentary/Perspective submissions do not require a peer review process, and Editor approval should be sufficient to proceed with its publication. However, since the editorial office approached me to review this work, I wish to recommend that this submission be considered for publication as either a Commentary or Perspective paper, depending on the journal's format.
Author Response
Thank you for your thoughtful review and feedback on our submission. We appreciate your recognition of the nature of our piece as a Commentary/Perspective on the usage of mpMRI in prostate cancer management. We agree with your suggestion to consider our submission for publication in either category, depending on the format of the journal.
Reviewer 3 Report
Comments and Suggestions for Authors
"Interpreting Prostate MRI Reports in the Era of Increasing Pros- 2 tate MRI Utilization: A Urologist’s Perspective"
In this literature review authors examined the immediate impact on healthcare, particularly focusing on the workload and evolving roles of radiologists and urologists tasked with the interpretation of these reports and consequential decisions regarding prostate biopsies.
With the help of mpMRI, Radiologists, can be able in divide patients who may benefit from more invasive treatment or exclude patients who may be harmed by overtreatment.
The demand for high-quality mpMRI, is estimated at approximately two million additional prostate MRI scans annually in Europe.
There is the need of a strong collaboration for effective reporting.
In the introduction could be added the reference doi: 10.1016/j.purol.2020.12.008. at line 14-15
Author Response
Dear reviewer,
Thank you for your helpful suggestion regarding our manuscript. We have added the reference (doi: 10.1016/j.purol.2020.12.008) to our introduction and it significantly enriches the depth of our references.